# CBLab: Scalable Traffic Simulation with Enriched Data Supporting

## Abstract

Traffic simulation provides interactive data for the optimization of traffic control policies. However, existing traffic simulators are limited by their lack of scalability and shortage in input data, which prevents them from generating interactive data from traffic simulation in the scenarios of real large-scale city road networks.

In this paper, we present **City Brain Lab**, a toolkit for scalable traffic simulation. CBLab consists of three components: CBEngine, CBData, and CBScenario. CBEngine is a highly efficient simulator supporting large-scale traffic simulation. CBData includes a traffic dataset with road network data of 100 cities all around the world. We also develop a pipeline to conduct a one-click transformation from raw road networks to input data of our traffic simulation. Combining CBEngine and CBData allows researchers to run scalable traffic simulations in the road network of real large-scale cities. Based on that, CBScenario implements an interactive environment and several baseline methods for two scenarios of traffic control policies respectively, with which traffic control policies adaptable for large-scale urban traffic can be trained and tuned. To the best of our knowledge, CBLab is the first infrastructure supporting traffic policy optimization in large-scale urban scenarios. The code is available on GitHub: `https://github.com/CityBrainLab/CityBrainLab.git`.

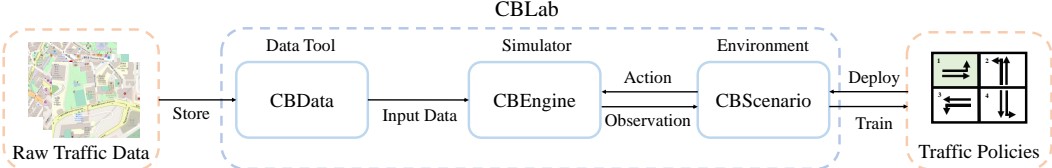

Figure 1: An overview of CBLab.

## 1 Introduction

Well-crafted traffic control policies, such as traffic signal control and congestion pricing, are expected to improve the efficiency of urban transportation. In recent years, many studies have been conducted to optimize the traffic control policies according to real-time traffic data Haarnoja et al. (2018); Wei et al. (2018; 2019a); Zheng et al. (2019); Zhang et al. (2020); Chen et al. (2020a); Zang et al. (2020); Oroojlooy et al. (2020); Wu et al. (2021); Zhang et al. (2022); Mirzaei et al. (2018); Qiu et al. (2019). These policies depend on data generated by interaction with the traffic environment where they explore to make good decisions under different consequences.

However, real-world urban traffic cannot provide enough interactive data to train these policies, because the exploration of the policy may have a toxic impact on the urban traffic e.g. provoke severe congestion. Traffic simulators are therefore born as alternatives to provide traffic environments for traffic control policies to interact with. These simulators Lopez et al. (2018); Zhang et al. (2019); Chen et al. (2020b) simulate the microscopic evolution of the urban traffic. For each time step, they describe the traffic state, obtain a traffic action from the decision of traffic control policies and make it happen in the simulation. traffic control policies can then learn from how the traffic evolves under certain actions and improve decision making.

While existing traffic simulators help hatch various traffic control policies successfully, they still come with drawbacks. Current simulators, as they were designed primitively, support simulation in road networks smaller than one hundred intersections and cannot scale to city-level traffic, which involves thousands of intersections. Due to limits in efficiency and mechanisms, these simulators are either not able to conduct a city-level simulation in a feasible time or set to prevent masses of vehicles from coming in the traffic.

Another concern lies in the shortage of input data for large-scale traffic simulation. Although the map data of main cities in the world is now completed roughly and being refined, there is an absence of infrastructure for convenient access to the map data and a pipeline to transform it into simulation inputs. Therefore, inputs for traffic simulation only come from manual work and are limited to a small set of road networks Wei et al. (2019c;b); Zheng et al. (2019); Xu et al. (2021) whose scales are often dozens of intersections (e.g. 4x3 or 4x4) - much less than real urban road networks.

To overcome two aforementioned drawbacks, we propose **C**ity **B**rain **Lab**, a novel toolkit for scalable traffic simulation. CBLab consists of three components: a microscopic traffic simulator CBEngine, a data tool CBData, and a traffic control policy environment CBScenario. CBEngine is of high efficiency which benefits from well-designed parallelization. With ordinary computing hardware, CBEngine is capable of running the traffic simulation on the scale of 10,000 intersections and 100,000 vehicles with a real-simulation time ratio of 1:4. CBData includes an accessible dataset that contains raw road networks of 100 main cities all around the world. A pipeline is prepared to automatically transform the raw data into input data for traffic simulation. Combining CBEngine and CBData, users can easily start up traffic simulation on real city-level road networks. Based on the scalable traffic simulation, we further implement CBScenario as an environment for two common traffic control policies: traffic signal control and congestion pricing. Users can design, develop, and train traffic control policies in the framework of CBScenario. To the best of our knowledge, we are the first to provide infrastructure for large-scale traffic control policy optimization.

Our contribution can be summarized as follows.

- We develop a scalable traffic simulator CBEngine which supports city-level microscopic traffic simulation for the first time.
- We develop a data tool CBData to provide input data for large-scale traffic simulation.
- Based on CBEngine and CBData, we implement an interactive environment CBScenario for two common traffic control policies under a large-scale setting.

## 2 CBENGINE: CITY-SCALE TRAFFIC SIMULATION ENGINE

In this section, we introduce the design of CBEngine. We demonstrate the modeling of urban traffic and conduct extensive experiments to show the efficiency and scalability of CBEngine.

### 2.1 THE TRAFFIC MODEL OF CBENGINE

The objective of the simulator is to describe the interaction between the road network and the traffic flow (vehicles in the traffic). As shown in Figure 2, the road network involves the interaction through one of its components: the traffic signal lights, which control the passing of vehicles at intersections. When the simulation starts, vehicles in the traffic flow set out from their origins, travel down the routes, and finally arrive at their destinations.

Specifically, traffic signals and vehicles interact as follows. Each traffic signal may change the signal phase (controlling the traffic direction allowed to move) as time changes. Vehicles move on, accelerating or decelerating according to the current speed, the traffic signal the movement of nearby vehicles, and other circumstantial factors. Meanwhile, they may also change their routes accordingly. For each time step, traffic signals, and vehicles observe from their views and make decisions on their next action $a_t$ (driving and routing). The simulator will then conduct these actions. This forwards the system from state $s_t$ at time $t$ to a new state $s_{t+\delta t}$ at time $t + \delta t$. One such step can be concluded by Eq 1. Due to the page limit, concrete modeling of the road network, traffic signals and traffic flows is discussed in Appendix C.

$$s_t + a_t \rightarrow s_{t+\delta t} \tag{1}$$

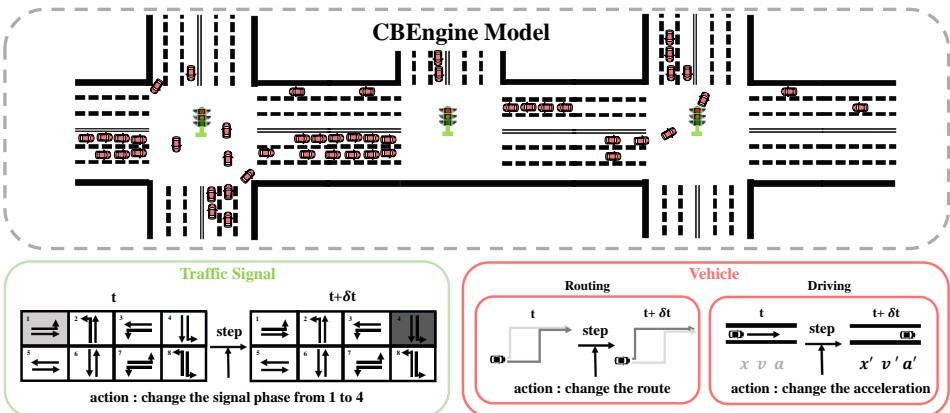

Figure 2: The Traffic Model of CBEngine

## 2.2 Vehicle Models: Driving and Routing

Behaviors of vehicles are controlled by vehicle models in the traffic simulation. Towards simulating the behaviors of vehicles, researchers have proposed various models Krauß (1998); Yuan et al. (2010; 2011). These models can be summarized into two categories: driving models and routing models. Driving models determine how vehicles move on the road while routing models make choices on routes.

Following Cityflow Zhang et al. (2019), a widely-used traffic simulator, the default driving model of CBEngine is a modification version of the model proposed by Stefan Krauß Krauß (1998). The key idea of this model is to drive as fast as possible subject to safety regularization, with nearby vehicles, traffic signals, and cutting-in considered. Due to the page limit, we discuss the details of the driving model in Appendix C.

An important function provided in CBEngine is vehicle model customization. In the simulator, both the routing model and the driving model are black boxes with definite inputs and outputs. Hence, we modularize two models as an independent C++ class in our implementation, respectively. Users can customize two models by only editing the class without modification to other parts of CBEngine. Our documentation in Appendix A provides detailed instructions. To the best of our knowledge, CBEngine is the first traffic simulator to support easy-to-use customization of vehicle models.

## 2.3 Efficiency and Scalability

The major bottleneck of supporting city-wide traffic control policy training is the limit in efficiency and scalability of existing simulators, of which extensive efforts have been made in CBEngine for improvement. In the following part, we illustrate a design adapted by CBEngine that greatly promotes the efficiency of CBEngine. Then, we conduct extensive experiments to validate the efficiency and scalability of CBEngine. More implementation details of CBEngine are discussed in Appendix D.

**Parallel Design in Computing the Vehicle Behavior**  The future state in the simulation is determined by the present state and the action. Therefore, computing the action is the main computational job. In CBEngine, we carefully design the parallelization process of computing the action for vehicles. The comparison of our design and that of other simulators is shown in Figure 3.

The process to compute actions for vehicles can be divided into two stages: getting the Status and getting the Action. In the first stage, the simulator collects information for computing the action of a vehicle. The set of information is denoted as the *Status*. In the second stage, the simulator computes the action according to Status. Existing simulators parallelize these two stages respectively. Vehicle objects in the simulator need to keep two data members: the Status and the Action. Two members are updated sequentially by two parallelized methods: GetVehicleStatus() and GetVehicleAction().

In CBEngine, we assemble two stages in one parallelized method: Compute(). We implement this by adjusting data dependencies and reconstructing the parallel architecture. This design benefits

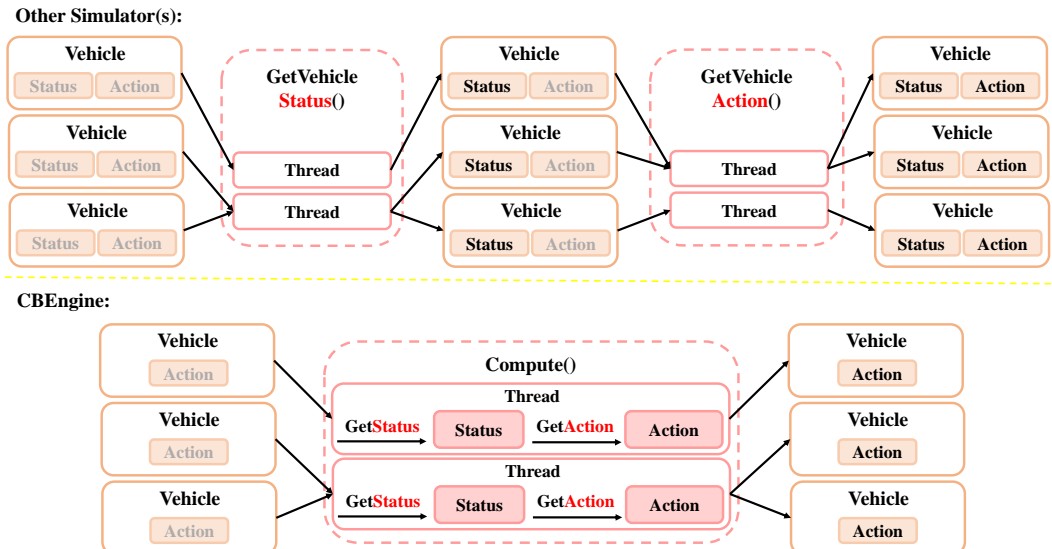

Figure 3: Comparison between CBEngine and other simulators on parallelization design of computing actions for vehicles. The gray data member indicates it is not updated, while the black one is updated.

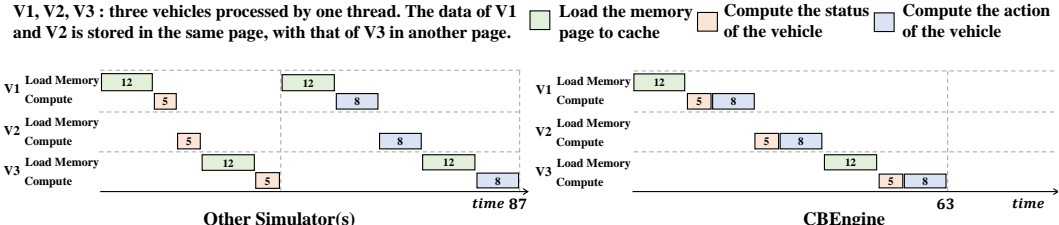

Figure 4: A schedule case of computing vehicle action. The length of the bar indicates the time the corresponding task consumes.

efficiency from two perspectives. First, the space cost is lower. We bind the Status data on the Thread object rather than on the Vehicle objects. This is because the number of threads is quite smaller than that of vehicles. Second, by merging two stages, the CPU is more likely to access existing pages in the memory when getting the action, because those pages are recently loaded to the memory when getting the status. Since this design reduces the number of times that the CPU loads pages from the memory, it decreases the number of cache misses and the time cost.

Figure 4 raises a case comparing the scheduling of one thread processing vehicles in other simulators to that in CBEngine. For each vehicle, the thread needs to compute its status and then its action accordingly. The page where the vehicle is stored is required to be loaded in memory for access to the vehicle. Assume that the cache can only store one page. For other simulators, the processing is divided into two stages, while each stage loads two pages from memory. This is because the thread needs to access all three vehicles in each stage. By contrast, processing in CBEngine combines two stages and does not access new vehicles until the job on the vehicle is finished. This design helps reduce the operation of loading pages and saves processing time.

**Experimental Setup** To evaluate the efficiency and scalability of our simulator, we compare it with two widely-used open-source microscopic traffic simulators, SUMO Lopez et al. (2018) and Cityflow Zhang et al. (2019). We compare these simulators in three aspects, running time, road network scalability, and traffic flow scalability. All the experiments are conducted on a Ubuntu20.04 system with a 40-core CPU and 128GB RAM. The unit of time cost is second for all three experiments. More details of the experimental setup are given in Appendix B.

**Experiment 1: Efficiency** We run a one-hour traffic simulation on urban traffic cases of six cities with distinct scales and compare the time cost of baselines and our simulator. Road networks of

| Dataset | Nanchang | Changchun | JiNan | Shenzhen | Hangzhou | Shanghai |
|---|---|---|---|---|---|---|
| Intersection Num | 1506 | 2228 | 2314 | 3427 | 3434 | 8474 |
| Vehicle Num | ~25000 | ~50000 | ~50000 | ~70000 | ~70000 | ~120000 |
| | Time Cost | | | | | |
| SUMO | 1239.93 ($\pm$3.58) | 2091.60 ($\pm$7.84) | 2151.01 ($\pm$70.64) | 3103.58 ($\pm$110.08) | 3199.14 ($\pm$70.87) | 6173.51 ($\pm$75.27) |
| Cityflow | 164.08 ($\pm$6.30) | 243.22 ($\pm$11.85) | 242.47 ($\pm$21.34) | 289.31 ($\pm$1.74) | 310.98 ($\pm$18.22) | 664.18 ($\pm$21.01) |
| CBEngine | **104.39** ($\pm$**0.54**) | **169.36** ($\pm$**1.60**) | **174.29** ($\pm$**1.93**) | **243.26** ($\pm$**1.75**) | **245.01** ($\pm$**2.16**) | **472.07** ($\pm$**2.90**) |

Table 1: Comparison results of efficiency experiments in six real-world cities.

these cases are obtained and cleaned from OpenStreetMap Haklay & Weber (2008). Traffic flows are generated according to the scale of the road network. Results are displayed in Table 1.

We can observe that CBEngine achieves significant improvement in simulation efficiency (usually 30%-40% compared with Cityflow and more than 90% compared with SUMO). The stability of CBEngine is distinctly better than that of baselines. Furthermore, the gap between baselines and ours grows with the scale of traffic cases, indicating that CBEngine can adapt well to large-scale cases.

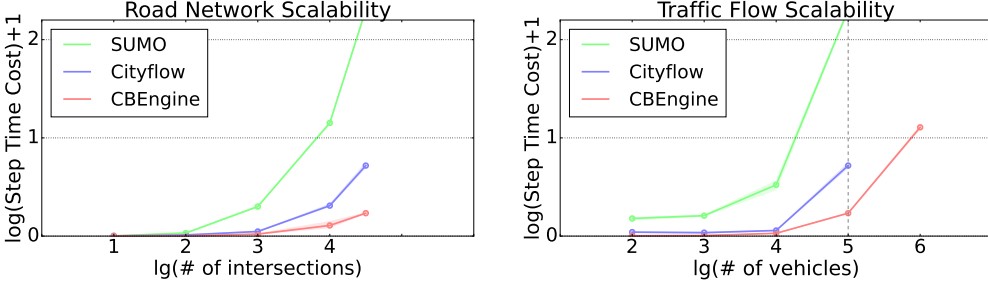

Figure 5: Comparison results of two experiments of scalability.

**Experiment 2: Scalability on Road Networks**    Simulators with high scalability on road networks can run the simulation efficiently on large-scale road networks. To explore the scalability of baselines and CBEngine on road networks, we run a traffic simulation on road networks of different scales. We select five regions from real road networks with $\{10, 10^2, 10^3, 10^4, 10^{4.5}\}$ intersections. The upper bound is set as $10^{4.5}$ because this is the largest road network for a single city in OpenStreetMap. For each setting, we repeat the experiment three times. We report the time cost of single-step simulation for baselines and CBEngine as well as the range.

The results are visualized in Figure 5 (left). CBEngine outperforms two baselines in time cost on road networks with all scales selected. Take the experiment setup under the road network with the largest scale as a quantitative example. The average single step time cost of CBEngine is 0.2670 seconds, while that of SUMO Lopez et al. (2018) and Cityflow Zhang et al. (2019) are 9.1832 seconds and 1.0343 seconds, respectively.

**Experiment 3: Scalability on Traffic Flows**    With high scalability on traffic flows, the simulator keeps efficient under heavy traffic. Similar to Experiment 2, we conduct an experiment to evaluate the scalability of traffic flows of baselines and CBEngine. We generate five traffic flows with $\{10^2, 10^3, 10^4, 10^5, 10^6\}$ on-way vehicles. For each setting, we repeat the experiment three times. We report the time cost of a single step for baselines and our simulator as well as the range.

The results are visualized in Figure 5 (right). CBEngine outperforms two baselines under different scales of traffic flows. To give a quantitative example, the average single step time cost of CBEngine under the traffic flow of $10^5$ vehicles is 0.2610 seconds, while that of SUMO Lopez et al. (2018) and

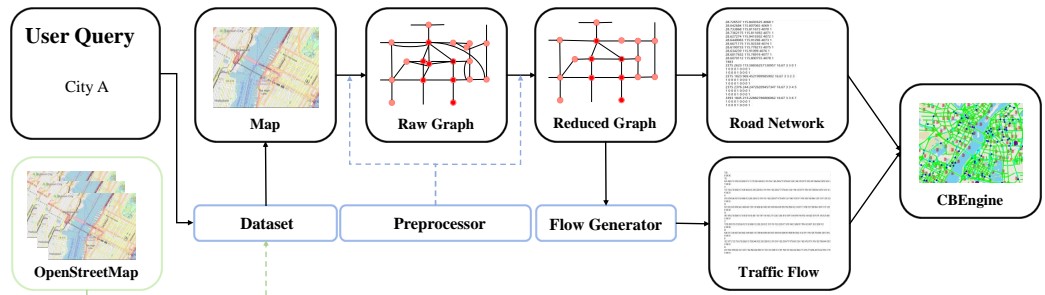

Figure 6: The simulation input data supporting pipeline of CBData.

Cityflow Zhang et al. (2019) are 8.9111 seconds and 1.1058 seconds, respectively. Specifically, two baselines are not able to run the case with 1,000,000 vehicles. We discuss the reason in Appendix D.

# 3 CBDATA: TRAFFIC DATA NETWORK CONNECTED TO CBENGINE

In this section, we introduce our data tool CBData. CBData serves to provide enriched input data supporting large-scale traffic simulation. The support is achieved by a dataset with raw road networks of 100 main cities all around the world and the transformation pipeline shown in Figure 6. Moreover, CBData includes two other pipelines that help the simulator learn from other traffic data.

## 3.1 PIPELINE: SIMULATION INPUT DATA SUPPORTING

Despite existing traffic simulators being capable of simulating the evolution of urban traffic, the application of traffic simulation is vastly limited by the shortage of input data. Specifically, to start up a simulation, the simulator takes road networks and traffic flows as input data. At present, there is no convenient access to these two kinds of data, although road networks of almost all main cities in the world can be extracted from open source map data Haklay & Weber (2008).

To disentangle this problem, we implement a pipeline to bridge open source map data and input data for simulation. This pipeline provides a one-click service to offer enriched input data for large-scale traffic simulation, solving the shortage of input data.

As shown in Figure 6, the pipeline consists of a dataset, a preprocessor, and a flow generator.

**Dataset:** We obtain the map data of the whole world from OpenStreetMap Haklay & Weber (2008). We extract the road network data of 100 main cities and store the data in our dataset. The dataset is now available on Google Drive (See Appendix A). Users can directly download the data and pick up their interested road networks. Details of the dataset are given in Appendix E.

**Preprocessor:** The preprocessor first constructs the road network as a raw graph by matching and connecting edges and nodes. The raw graph is then cleared to remove the redundant nodes and graphs. This is necessary because redundancy is common in open-source map data. After removing the redundancy, the reduced graph is transformed into the road network in the standard format.

**Flow Generator:** The flow generator generates the traffic flow for a road network. Given the total number of vehicles, the generator assigns origins and destinations for these vehicles, respectively, which distribute as averagely as possible. The default route for each pair of the origin and the destination is the shortest path. However, the route can be changed by the routing model of the traffic simulator when running the simulation.

## 3.2 PIPELINE: LEARNING TO SIMULATE FROM TRAFFIC DATA

In addition to map data, there is a lot of other traffic data with the potential to enhance the plausibility of traffic simulation. In CBData, we propose two paradigmatic pipelines to illustrate how to learn to simulate from traffic data. Note that we are not to propose effective methods but provide a paradigm for learning to simulate.

### 3.2.1 Learning to Simulate Driving

The driving model determines how drivers accelerate and decelerate according to their observation of the circumstance. The driving behavior in different traffic or different cities can be distinctly different. Therefore, learning the driving parameters of the traffic simulator from the traffic data is sound for traffic simulation. It helps the simulator to behave more plausibly like the local drivers.

The goal of learning to drive is to find a set of driving model parameters that can minimize the gap between the traffic data and the simulator, *e.g.* that between the observed speed in the real data and the observed speed in the simulator, with the same traffic flow. As mentioned in Section 2.1, the driving model of CBEngine is easy to modify. Here, we adopt the default model (See Appendix C for more details) and select three parameters from the model as the parameters to be optimized: acceleration maximum, deceleration maximum, and speed limit.

We use a black-box optimization toolkit OpenBox Li et al. (2021) as the optimization tool. Note that users can use any other optimization toolkit according to their needs. OpenBox searches for parameters to fill the gap between simulation observation and the ground truth. The 1-hour GPS trajectory data of vehicles on two roads in Shenzhen, China are used to search the parameters. The observation interval is 1 minute. For every 20 minutes, the traffic distribution will change. We expect that acceleration parameters can be continuously learned when the data changes.

The loss curves of the learning process of these two avenues are displayed in Figure 7. The gap between the observed speed average and the ground truth is decreasing as time changes. After the traffic changes at the 20th and the 40th minute, the driving model can not fit the new traffic data. Hence, we observe a high loss immediately. After a few minutes, the gap continuously decreases. Note that, the difference still maintains a certain positive value. This implies the potential to change the driving module to pursue a better performance of driving module correction.

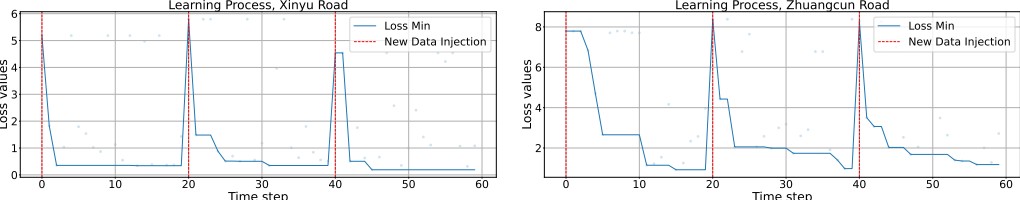

Figure 7: Loss curves during learning the driving module on two roads.

### 3.2.2 Learning to Simulate Routing

The routing model determines how vehicles route themselves, given the origin and destination. A data-dependent routing model can be formulated as a route generator, which generates a route for certain origin and destination based on real trajectories. However, the studies in this field are limited and we exploit the Recurrent Neural Networks (RNNs) as our route generator Choi et al. (2021).

We use part of a vehicle trajectory dataset in Shenzhen, China. This dataset contains 22 different routes in total. We train the RNN Choi et al. (2021) to learn the distribution of routes and conduct routing for vehicles in the simulator. Figure 8 shows the result of loss curves and two trajectory generating metrics, BLEU Papineni et al. (2002) and METEOR Banerjee & Lavie (2005). The loss converges to 0 at the first twenty iterations, while BLEU and METEOR get close to 1. This indicates that at this stage, routes generated overlap at least one route in the real trajectory data.

In addition, the routing result is visualized in Figure 9. Compared with an untrained generator, the trained generator recognizes the main distribution pattern of real trajectories. Meanwhile, it tends to ignore some infrequently-appearing routes.

## 4 CBSCENARIO: ENVIRONMENT FOR LARGE-SCALE TRAFFIC CONTROL POLICIES

In this section, we introduce CBScenario, an interactive environment for large-scale traffic control policies. CBScenario benefits from the large-scale traffic simulation supported by CBEngine and

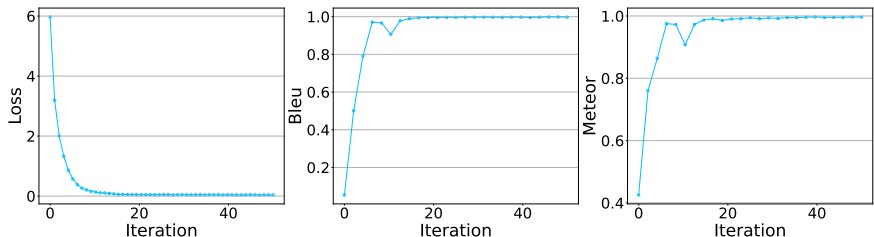

Figure 8: Curves of loss, BLEU, and METEOR during learning the routing module.

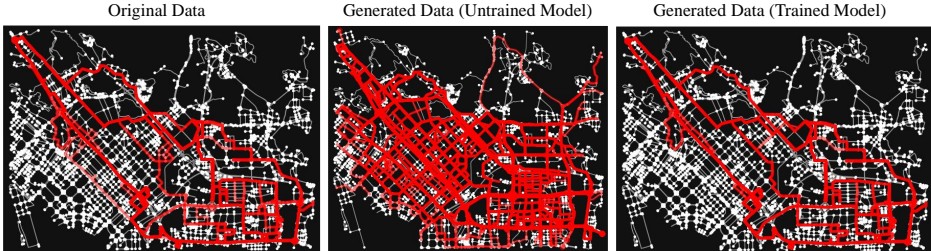

Figure 9: Visualization of the routing result. The deeper red is, the more frequent the route is picked.

CBData and is capable of training traffic control policies for city-level traffic. Concretely, CBScenario includes benchmarks for two traffic control policies: "Traffic Signal Control" and "Congestion Pricing". We conduct experiments on our environment to show the plausibility of the traffic simulation.

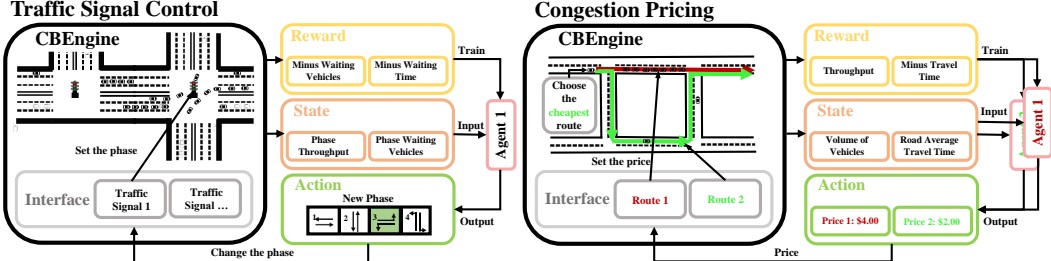

Figure 10: Illustration of two scenarios: traffic signal control and congestion pricing.

## 4.1 TRAFFIC CONTROL POLICY 1: TRAFFIC SIGNAL CONTROL

The traffic signal control problem Wei et al. (2019c) tries to improve the performance of urban traffic by carefully choosing the phase of traffic signals at intersections. An ideal traffic signal control policy can capture the global and local traffic dynamics and allocate more passing time to the phase with higher traffic pressure. Figure 10 (left) shows the problem setting of traffic signal control.

In consideration of the Markov nature of traffic signal control, the traffic signal control can be formulated as a Markov Decision Process (MDP):

- **State:** Intersection-level and road-level observation and statistics of observation, *e.g.* the number of waiting vehicles on the road, historical average vehicle throughput of different phases at the intersection.
- **Action:** Decide which directions can pass.
- **Reward:** Metrics measuring the performance of urban traffic. We provide two widely-used metrics: the total number of waiting vehicles at the intersections and the average waiting time during an action interval.

We implement several baseline algorithms to justify the plausibility of our simulation. We consider two metrics to evaluate the performance of these algorithms: arriving vehicle throughput and average travel time of vehicles. More details of the experiment are given in Appendix B. The results are

shown in Table 2. Two transportation methods, MaxPressure Varaiya (2013) and Self Organized Traffic Light (SOTL) Cools et al. (2013), show a degree of advantages in increasing throughput and reducing travel time. However, as a learning based traffic control policy, DQN Mnih et al. (2015); Wei et al. (2018) performs even better.

| Dataset | Hangzhou | | Manhattan | |
|---|---|---|---|---|
| Metrics | Throughput | Travel Time(s) | Throughput | Travel Time(s) |
| FixedTime | 2184 | 1478.01 | 2894 | 1309.88 |
| MaxPressure | 3336 | 700.66 | 3364 | 805.14 |
| SOTL | 1122 | 305.79 | 137 | 488.62 |
| DQN | 3573 | 309.10 | 3926 | 375.51 |

Table 2: Performance of baseline algorithms on traffic signal control.

### 4.2 TRAFFIC CONTROL POLICY 2: CONGESTION PRICING

Congestion pricing reroutes vehicles by dynamically assigning prices to different routes and guiding vehicles to drive on the route of the lowest price. Good pricing methods tend to allocate heavy traffic on different routes with a trade-off of the distance and the capacity, therefore enhancing traffic efficiency. Figure 10 (right) shows the setting of congestion pricing.

Similar to traffic signal control, we can define congestion pricing as an MDP:

- **State:** Road-level observation and statistics of observation. For the observation common used, we have the vehicle number and the average speed of vehicles on the road.
- **Action:** Price roads in the road network.
- **Reward:** Metrics measuring the performance of urban traffic. A common reward is the average travel distance of vehicles during an action interval.

Congestion pricing has been researched in the transportation field for a while yet few studies use data-driven methods. We implement two transportation-based methods, Random and Deltatoll Sharon et al. (2017), and an RL algorithm, EBGtoll Qiu et al. (2019). More details of the experiment are given in Appendix B. Results are shown in Table 3. Random outperforms No-change which keeps original routes for vehicles. This is plausible because random rerouting allocates traffic averagely on all available routes. Deltatoll and EBGtoll behave similarly and outperform Random in both evaluation metrics.

| Dataset | Hangzhou | | Manhattan | |
|---|---|---|---|---|
| Metrics | Throughput | Travel Time(s) | Throughput | Travel Time(s) |
| No-Change | 2176 | 1455 | 2911 | 1328 |
| Random | 3008 | 644.00 | 3459 | 1120.69 |
| Deltatoll | 3186 | 604.00 | 3670 | 960.17 |
| EBGtoll | 2803 | 310.18 | 3494 | 1019.85 |

Table 3: Performance of baseline algorithms on congestion pricing.

## 5 CONCLUSION

In this paper, we present CBLab, a toolkit for scalable traffic simulation. CBLab provides the first simulator to support real-time simulation of large-scale cities with more than 10,000 intersections. A data tool is implemented to supply large-scale input data for the simulation. An interactive environment and benchmarks for two common traffic policies have been built and prove that the simulation is plausible. To the best of our knowledge, CBLab is the first infrastructure supporting the training of large-scale traffic policies.

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

# A    KEY INFORMATION OF CBLAB

## A.1    LICENSING

CBLab uses the MIT license.

## A.2    CODE

The code of CBLab is available on GitHub.
`https://github.com/CityBrainLab/CityBrainLab.git`

## A.3    DOCUMENTATION

The documentation of CBLab is available.
`https://cblab-documentation.readthedocs.io/en/latest/`

## A.4    ROAD NETWORK DATASET

The road network dataset is available on Google Drive:
`https://drive.google.com/drive/folders/1IyTvWprOA1R_`
`6PVkuh7v9R4xrHcZAmYT?usp=sharing`

## A.5    LIMITATIONS & FUTURE WORKS

CBLab aims to provide efficient and data-driven traffic simulation for research in traffic policies. Although we provide interfaces for using data to optimize the simulator, we cannot obtain all real traffic data from different cities to conduct complete optimization on our simulator. We will invite future contributors to make more traffic data compatible with these interfaces so that they can be used to optimize the simulator and run the simulation.

Moreover, we plan to implement more scenarios for traffic policies, *e.g.* traffic restriction. Also, we aim to include more powerful algorithms in these scenarios, *e.g.* Intellilight and FRAP for traffic signal control. It is our hope that CBLab can serve as an online benchmark for various scenarios with state-of-the-art algorithms in addition to supporting powerful traffic simulation.

## A.6    POTENTIAL NEGATIVE SOCIAL IMPACTS

All traffic simulators will suffer from the gap between simulated and simulated observations. This gap may lead to biased observations and traffic policies based on the simulation. CBLab releases the first traffic simulator open for users to optimize so that it tends to perform as the real traffic. However, the traffic patterns from different places are different. It is worth considering how to ensure the reality of our traffic simulation when simulating the traffic from different places. Hence, we are collecting more traffic data and studying the common patterns among the data, with the hope of trying to resolve this problem.

# B    DETAILS OF EXPERIMENTAL SETUP

## B.1    EFFICIENCY AND SCALABILITY (SECTION 2.3)

**Baselines Setup**    In our experiment, we use the default setting of SUMO (TRACI) and Cityflow. Note that our simulator is a microscopic one. Therefore, we select two open source microscopic traffic simulators as our baselines. The routing model is not activated for all three simulators so it is not relevant if routing parallelization is used. In the experiment presented in our paper, internal links between intersections are used.

Considering that CBEngine simplifies links between intersections, a question may be raised whether the internal link has a decisively negative impact on the performance of the baseline. To answer this question, we conduct new efficiency experiments on SUMO with no internal links and the same setting otherwise. The comparison result is demonstrated in Table 4.

| Dataset | Nanchang | Changchun | JiNan | Shenzhen | Hangzhou | Shanghai |
|---------|----------|-----------|-------|----------|----------|----------|
| | Time Cost | | | | | |
| SUMO (with internal links) | 1239.93 ($\pm$3.58) | 2091.60 ($\pm$7.84) | 2151.01 ($\pm$70.64) | 3103.58 ($\pm$110.08) | 3199.14 ($\pm$70.87) | 6173.51 ($\pm$75.27) |
| SUMO (no internal links) | 1218.53 | 1987.12 | 2046.63 | 2973.50 | 3020.16 | 6083.37 |

Table 4: Comparison results of efficiency experiments between two setups of SUMO.

According to the result, removing internal links for SUMO does improve the simulation efficiency. However, SUMO's efficiency is still not considerable compared with that of CBEngine. This is because CBEngine deploys other optimization (e.g. an optimized parallelization architecture) to improve efficiency.

**Datasets** The efficiency experiment is conducted on the traffic data from six cities of different scales: Nanchang, Changchun, Jinan, Shenzhen, Hangzhou, and Shanghai. We obtain the road network data from our dataset of CBData. Traffic flows are generated according to the scale of the road network. Two scalability experiments are conducted on real-world road networks from our dataset with scales close to the selected road network size ($\{10, 10^2, 10^3, 10^4, 10^{4.5}\}$ intersections). Traffic flows are generated according to the scale of the road network. All road networks and traffic flows are available in our code provided on Google Drive.
```
https://drive.google.com/drive/folders/1e8wjEYFnDXluHaOxyAzJOOknNvJPZ4_
r?usp=sharing
```
We also provide a reproducing instruction to help reproduce our experimental results.
```
https://github.com/CityBrainLab/CityBrainLab/blob/main/CBEngine/
Reproducing_Instruction.md
```

**Computing Resources** All the experiments are conducted on a Ubuntu20.04 system with a 40-core CPU and 128GB RAM.

**Hyperparameters** The number of threads is chosen as 20 to stay consistent with the number of used cores. Note that, using fewer or more threads will lead to worse efficiency for both Cityflow and CBEngine.

**Error Bars** Error bars of the experiment are shown in Table 1 and Figure 3.

### B.2 LEARNING FROM REAL TRAFFIC DATA (SECTION 3.2)

In these experiments, we aim to provide demonstrations to show the possibility to use real-world traffic data to optimize the traffic simulator. Hence, there might be further room for improvement if the parameters are tuned carefully.

#### B.2.1 LEARNING TO SIMULATE DRIVING

**Datasets** The road network of Shenzhen is obtained from our dataset. We obtain the traffic flow (as the input) and the observation of speed (as the ground truth) from the GPS trajectory data of cars in Shenzhen, China for one day. The data covers 123,481 trajectories and comes from personal data providers with consensus. We are working on releasing a processed version by removing sensitive data.

**Optimization Details** We use OpenBox as a toolkit to search for parameters. The code can be found at `https://github.com/PKU-DAIR/open-box.git` under the MIT license. The start-up hyperparameters are as follows. For the maximum acceleration and deceleration, we set the default value (value where the search starts) as $2.0m/s^2$ and $5.0m/s^2$, respectively. For the speed

limit, we set the default value as $11.1m/s$. The number of rounds is set as 20. We use the surrogate type *auto* and optimizer type *auto*.

### B.2.2 LEARNING TO SIMULATE ROUTING

**Datasets** We demonstrate how to learn the routing module on trajectories data of Shenzhen, China for one day. We collect the trajectories with the origin of {Latitude: $22.5405°N$, Longitude: $113.967°E$} and the destination of {Latitude: $22.6164°N$, Longitude: $113.853°E$}. The origin is the bus station of the Window of the World, a famous scenic spot in Shenzhen. The destination is a bus station on the highway from Shenzhen to Guangzhou. This origin-destination pair aggregates the most number of different routes in our dataset, while routes of other origin-destination pairs are quite unified. The total number of different routes is 22 and that of trajectories is 118.

**Optimization Details** We use an RNN-based model as the trajectory generator. The code can be found at `https://github.com/benchoi93/TrajGAIL.git` under MIT license. We follow the setting of hyperparameters in the original paper Choi et al. (2021) except for the iteration number since our trajectories are complicated for the generator to learn from. We set the iteration number as 100.

### B.3 TRAFFIC SIGNAL CONTROL AND CONGESTION PRICING (SECTION 4.2 AND 4.3)

The goal of these two experiments is to provide possible benchmarks for algorithms. Here, we only use several typical algorithms to validate these scenarios. Providing comprehensive baseline methods comparison is out of the scope here. Hence, people are welcome to provide more advanced algorithms for these scenarios.

**Datasets** We use two real-world datasets to validate CBScenario: Hangzhou and Manhattan. Both datasets are transformed from the traffic data at `https://traffic-signal-control.github.io/#open-datasets`, which serves as a widely used benchmark for traffic signal control. We use part of CBData to transform them to the format suitble for CBEngine. The goal of the experiment is to validate the plausibility of our traffic simulation. Therefore, we refer to the widely used benchmark rather than picking up novel cases not being evaluated yet.

**Hyperparameters of Traffic Signal Control** The traffic in one episode lasts for 1800 seconds. The DQN method is trained for 50 episodes and the batch size is set as 64. Other hyperparameters are listed in Table 5.

| Hyperparameter | Memory size | Value network updating interval | $\epsilon$ | $\gamma$ |
|:---:|:---:|:---:|:---:|:---:|
| Value | 5000 | 1 | 0.9 | 0.95 |

| Hyperparameter | Learning rate | Target network updating interval | $\epsilon_{min}$ | Decay of $\epsilon_{min}$ |
|:---:|:---:|:---:|:---:|:---:|
| Value | 0.005 | 20 | 0.2 | 0.995 |

Table 5: Hyperparameters of DQN in traffic signal control.

**Hyperparameters of Congestion Pricing** The traffic in one episode lasts for 10800 seconds. Actions are taken every 540 seconds. We use the fixed time policy as the default traffic signal control policy. For transportation-based methods, we evaluate them in one episode. For the training of EBGtoll, the number of episodes is 200 and the batch size is set as 32. Other hyperparameters are listed in Table 6.

## C TRAFFIC MODELING OF CBENGINE

### C.1 ROAD NETWORK

In CBEngine, **Road Network** is the topological network where vehicles drives. It consists of two components: **roads** and **intersections**. **Road** models the road segment in the real-world road network.

| Hyperparameter | Memory size | Value network updating interval | Policy learning rate |
|:---:|:---:|:---:|:---:|
| Value | 2000 | 1 | 0.001 |

| Hyperparameter | $\tau$ | Target network updating interval | Critic learning rate |
|:---:|:---:|:---:|:---:|
| Value | 0.125 | 10 | 0.0005 |

Table 6: Hyperparameters of EBGtoll in congestion pricing.

A **road** may include multiple **lanes**. Each **lane** holds one or more vehicles. **Intersection** is the nexus of different **roads**. Via **lane links** in the **intersection**, **lanes** of different **roads** connect to each other. **Traffic signal light** is another key element in the **intersection**, which assigns a true-or-false signal for each **lane link**. Vehicles can only move from one road to another through available **lane links** with a true signal.

### C.1.1 MODEL DESIGN

The default driving model of CBEngine is a modified version of the driving model used in Cityflow, originating from the driving model proposed by Stefan Krauß. The key idea is that: the vehicle will drive as fast as possible subject to safety regularization. Specifically, the maximum speed of the vehicle is subject to several static or dynamic speed constraints. Vehicles will accelerate or decelerate to the speed with the maximum acceleration or deceleration. In the implementation, the maximum acceleration (deceleration) is a hyperparameter and is open to users to tune with an API. The considered speed constraints are listed and discussed respectively as follows:

- Road speed limit
- Collision-free following & leading speed
- Cutting-in collision-free speed
- Traffic-signal safe speed

**Road speed limit** Each road has its own speed limit. This is a static speed constraint.

**Collision-free following & leading speed** To avoid collisions, vehicles need to adapt their speed to the speed of their following and leading vehicles. We use the collision-free following speed to model the max speed constrained by the leading vehicle. Following the driving model in Cityflow, we compute these two constraints with Eq 2. It takes $v$ current speed of the vehicle, $v_L$ current speed of the leading vehicle, $d$ maximum deceleration of the vehicle, $d_L$ maximum deceleration of the leading vehicle, $D$ the current distance between two vehicles, $interval$ the length of each time step as parameters to compute the collision-free following speed $s_{cfs}$.

$$
\begin{aligned}
a &= \frac{1}{2 \cdot d} \\
b &= \frac{interval}{2} \\
c &= \frac{v \cdot interval}{2} - \frac{v_L^2}{2 \cdot d_L} - D \\
s_{cfs} &= \frac{-b + \sqrt{b^2 - 4ac}}{2 \cdot a}
\end{aligned}
\tag{2}
$$

To compute the collision-free leading speed $s_{cls}$, we use a mirror equation of Eq 2. We use the $d_F$ the maximum deceleration of the following vehicle and $v_F$ the speed of the following vehicle to replace $d_L$ and $v_L$ in the Eq 2. The collision-free following and leading speed constraints can be summarized as Eq 3.

$$
s_{cls} \leq v \leq s_{cfs}
\tag{3}
$$

**Cutting-in collision-free speed**   CBEngine supports self-adaptive lane changing within the road (Cutting-in). This asks for cutting-in collision-free following and leading speed, avoiding collisions with the leading and following vehicles in the target lane. We compute this constraint using Eq 2 with $v_L$, $d_L$, and $D$ given by the leading vehicle in the target lane and $v_F$, $d_F$, and $D$ given by the following vehicle in the target lane.

An exception happens when the vehicle needs to conduct an emergent cutting-in. This takes place when the vehicle is very close to the intersection but still in the wrong lane (e.g. The vehicle is in the go-straight lane but needs to turn left). On this occasion, the vehicle tries to stop to wait until it is able to change the lane. Therefore, the maximum speed constraint equals zero and the vehicle will decelerate to zero with its maximum deceleration.

**Traffic-signal safe speed**   Vehicles heading for an intersection are subject to two constraints determined by the traffic signal. First, the current speed can be decelerated to zero within the remained passing time of the traffic signal. Second, the driving distance cannot exceed the distance to the intersection, under the decelerating process defined in the first constraint.

### C.1.2   EFFECTIVENESS

Although CBEngine provides convenient driving model customization, which decreases the impact of the default driving model, we conduct two experiments to evaluate its effectiveness. First, we compare the average speed of vehicles in CBEngine and SUMO under the same simulation setups over 600 seconds. The experiments are finished on three of the experimental setups used in our efficiency experiments: Nanchang, Changchun, and JiNan. The result is visualized in Figure 11. Overall tendencies of two average speeds fit each other approximately, while that in CBEngine is more volatile. This can be explained by the difference in the driving model and will not have visible impacts on traffic flow statistics, which is the main factor in learning traffic control policies.

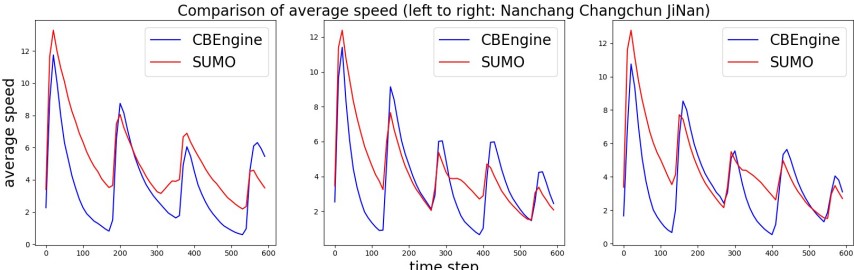

Figure 11: Comparison of the average speed of vehicles, CBEngine and SUMO

The second experiment focuses on the speed distributions of vehicles in CBEngine and SUMO. We compare the speed distributions in 150 seconds, 300 seconds, and 450 seconds under the experimental setup of Changchun used in our efficiency experiments. The comparison is visualized in Figure 12. Normalized Wasserstein distance between two distributions is shown in Table 7. The two distributions are similar to each other roughly but differ in details. The difference may not influence the overall performance of the simulated traffic in the level of traffic statistics. Also, the driving model customization supported by CBEngine makes it possible for users to use driving models according to their needs, which greatly improves the effectiveness of CBEngine, compared to SUMO and Cityflow.

| Time Step | 150 | 300 | 450 |
|---|---|---|---|
| Normalized Wasserstein Distance | 0.0141 | 0.0124 | 0.0089 |

Table 7: Normalized Wasserstein distance between speed distribution of CBEngine and SUMO

### C.2   APIS FOR TRAFFIC CONTROL

CBEngine provides various APIs for users to develop novel traffic control policies. The functions supported by these APIs are listed as follows:

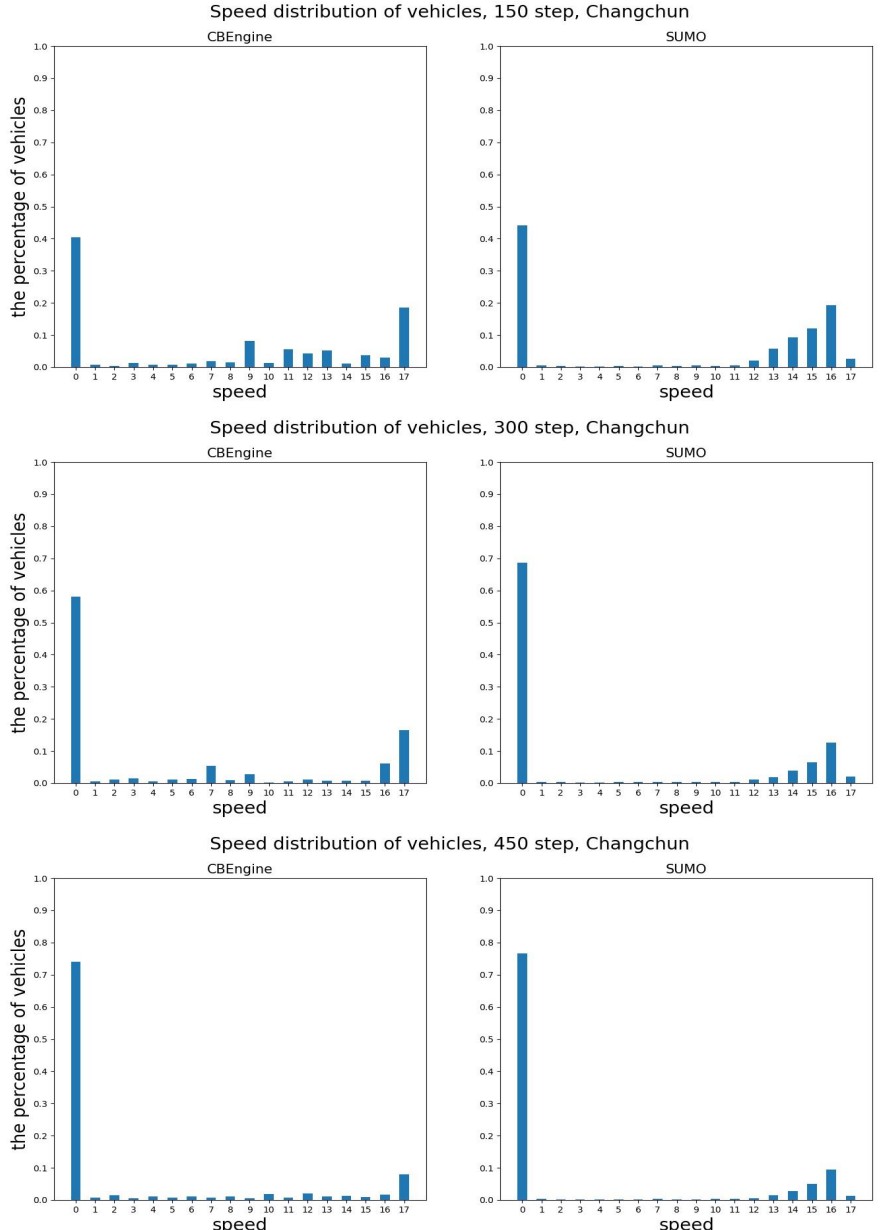

Figure 12: Comparison of speed distributions of vehicles, CBEngine and SUMO

- Changing the phase of a traffic signal light
- Modifying the speed limit of a road
- Changing the route of a vehicle

## D IMPLEMENTATION DETAILS OF CBENGINE

Except for the parallel design mentioned in Section 2.3, two mechanisms of CBEngine also contribute to its efficiency.

**Lane Changing in Driving Model** Lane changing is a driving action. Drivers may change the lane if they feel the current lane is too congested. However, lane changing is hard to simulate in the traffic simulation. Cityflow, one of our baselines, omits all lane changing except those happening at the

intersection. The lane of the vehicle is determined by the direction it will turn. This simplifies the implementation but leads to poor plausibility because vehicles on the road cannot make full use of all lanes as they do in real urban traffic. They have to stay at the current lane and may wait a long time, although their neighbor lane is clear. Furthermore, to avoid collision on this occasion, Cityflow does not allow new vehicles to come in until the lane is relatively unblocked. This limitation severely impacts the scalability of Cityflow and explains the fact in our experiment of scalability that Cityflow cannot hold 1,000,000 vehicles.

In CBEngine, we implement a driving model allowing lane changing. Vehicles are put in a random lane when they get into the road. They will try lane changing according to the direction they are to turn to. But if that lane is in congestion, they will keep going in the current lane which is relatively clear. This design achieves higher plausibility and provides stronger scalability for CBEngine.

**Intersection Links**   Another key mechanism of the traffic simulator is the intersection link. SUMO and Cityflow track the behavior of vehicles inside the intersection. However, due to the limit in the implementation, this design may lead to deadlocks very frequently in the practice, especially when running large-scale traffic simulations. This is because the track of some vehicles may block that of others. Therefore, we conduct simplification here to avoid such deadlocks. When a vehicle passes the intersection, the intersection will hold it for a while and then send it to the target road. We believe that the effect of the intersection link can be simulated by the holding time. We also discuss the impact of intersection links on our baselines in Appendix B. To the best of our knowledge, our design avoids all such deadlocks in practice.

## E   DETAILS OF ROAD NETWORK DATASET

The road network dataset in CBData includes raw road networks of 100 main cities around the world. The list and the range of these road networks are given on Github:
`https://github.com/CityBrainLab/CityBrainLab/blob/main/CBData/citylist.csv` We obtain the data from OpenStreetMap, an open-source map database. Note that bounding boxes of these road networks have to be a rectangle thus not strictly consistent with the boundary of the city. These road networks can be transformed into road network inputs for traffic simulation with our pipeline in CBData. Here, we visualize six road networks used in our experiment for example to give a scratch of our road network data in Figure 13.

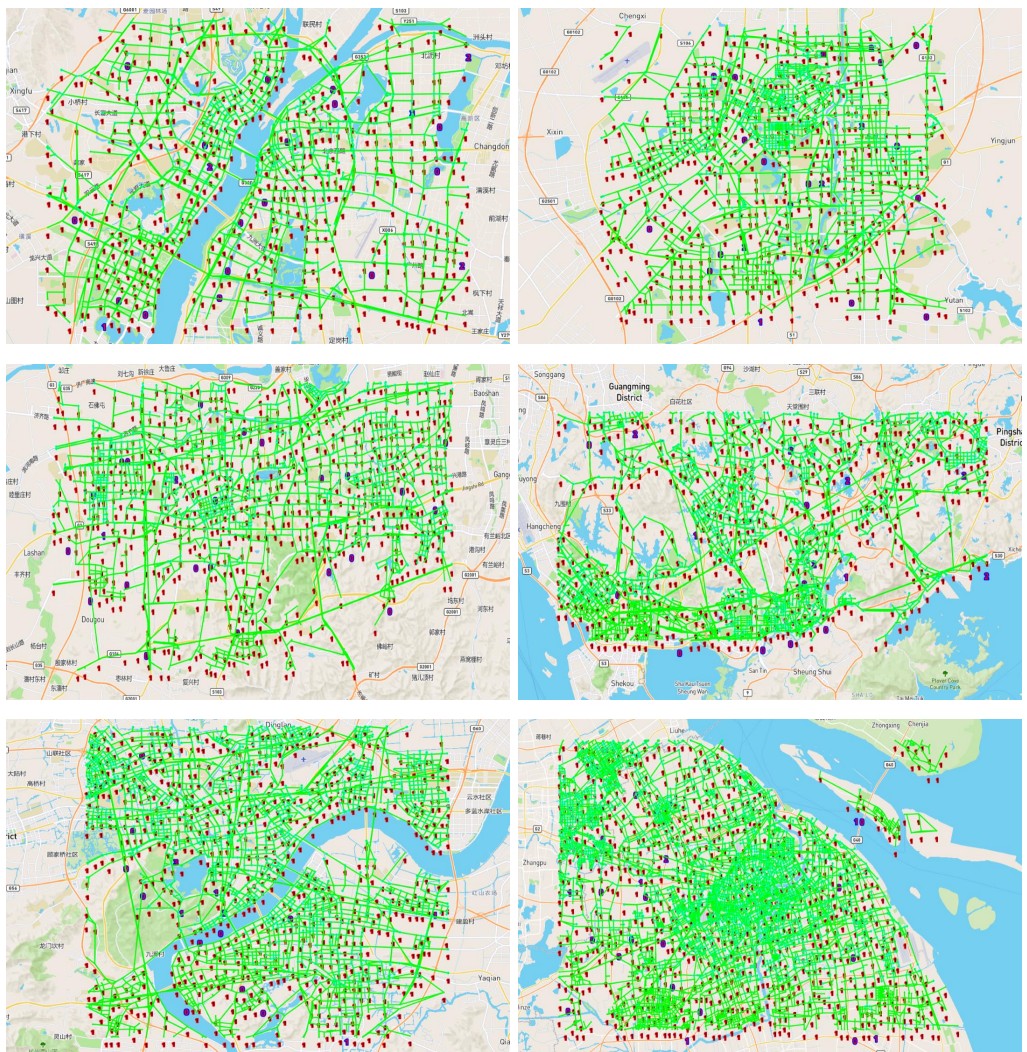

Figure 13: Visualization of road networks of six cities used in our experiment: Nanchang (top left), Changchun (top right), JiNan (middle left), Shenzhen (middle right), Hangzhou (bottom left), Shanghai (bottom right).

