# OpenReview forum: "CBLab: Scalable Traffic Simulation with Enriched Data Supporting"
_ICLR.cc/2023/Conference — Submitted to ICLR 2023_

### Official Review · Reviewer_JNCH · 2022-10-25

**Confidence:** 4
**Correctness:** 3
**Technical Novelty And Significance:** 3
**Empirical Novelty And Significance:** 3
**Recommendation:** 8

**Clarity, Quality, Novelty And Reproducibility:**

The paper is solid and with clear details. The code and document are attached at the end of the paper, which seem solid and reproducible.

**Strength And Weaknesses:**

This work proposes a novel simulator for urban transportation. The writing and logic are good. Some minor issues:
-	Some typos can be found in the manuscript, e.g. page 4: load -> loaded
-	For CBScenario, I wonder if the authors conduct traffic light control and congestion pricing with a singe agent RL or multi-agent RL, and if the agent merely controls a single intersection or road?


**Summary Of The Paper:**

This work is based on two concerns: 1. Current simulators cannot handle city-level simulation with thousands of intersections; 2. Current map data cannot be easily accessed and transformed into simulation inputs. In CBEngine, the authors design a paralleled module to improve computing efficiency. The experiments show that CBEngine has achieved significant progress compared to SOTA simulators. In CBData, the authors construct a pipeline to treat map data, and introduce new modules to adjust simulation parameters via traffic data. At last, two experiments are shown to provide the plausibility of the traffic simulation.

**Summary Of The Review:**

This work is interesting and meaningful, the writing is good and the overall paper is well-organized. I believe it can be helpful to both cs and transportation community to verify models or policies in a large scale.

---

> ### Author Response · Authors · 2022-11-16
> **Response to the review**
>
> We sincerely appreciate the insightful and encouraging review. It is especially precious for the field of learning traffic control policies, which makes sense to the public interests with little support. Also, we would carefully check the manuscript and make it out of typos.
>
> For the setting of traffic control policies in our experiments, we use single-agent RL because this only serves as a simple benchmark to validate the plausibility of our environment. Each intersection has its own agent and each agent makes decisions according to their local observation. However, our simulation supports both single-agent and multi-agent algorithms. Also, we would like to implement more baseline methods of both single-agent and multi-agent, to enrich our benchmarks. We also welcome and expect the new algorithm implementation from the open-source contribution.
>
> Again, we appreciate the helpful review and welcome any additional comments. Please feel free to contact us if you have any further questions.

---

### Official Review · Reviewer_VNgb · 2022-10-25

**Confidence:** 4
**Correctness:** 3
**Technical Novelty And Significance:** 3
**Empirical Novelty And Significance:** 3
**Recommendation:** 6

**Clarity, Quality, Novelty And Reproducibility:**

On page 2 authors say, "Due to limits in efficiency and mechanisms, these simulators are either not able to conduct a city-level simulation in a feasible time or set to prevent masses of vehicles from coming in the traffic". While later on, they show that the CityFlow is able to simulate all scenarios at just a fraction slower than the proposed framework.
Grammatical mistakes:
Abstract ->  CBLab is consist of three components: CBEngine, CBData, and CBScenario.
Page 4 -> because those pages are recently load to the memory when...
Page 4 ->The page where the vehicle is stored is required to be load in memory for access to
the vehicle.

There is discussion on parallelization in Section 2.3, but formal details are hidden behind the experimental results.
Page 5 -> "The stability of CBEngine is distinctly better than that of baselines" How the stability is calculated, is it a typo?
Page -> There is a discussion preprocessing, but how preprocessing is performed, the article is silent about it.

Generally, the paper is well-written and descriptive quality is good.

The code is provided upfront so the results can be easily verified.

**Strength And Weaknesses:**


The paper is well-written and organized in a manner that is easy to grasp. A thorough explanation of the experiments is provided. The graphs are used to show the results that are explained thoroughly. The paper is well structured and well organized.

Despite its importance, the paper seems to be an account of a development project that ended well. The experiments show the success of the framework and its interfaces, but there is not much fundamental scientific contribution. The framework will assist the scientists and will surely spur the research in the traffic analysis domain, but does the paper offer a new paradigm or a new algorithm? I am afraid the answer is no. The only contribution that can be associated with a conceptual level is the change in the architecture of the simulation engine that decreases page faults.

Even that is not significant as a reduction in page faults will only affect when the computer memory is low or the size of the simulation is very large. For all other cases, the change in architecture does not make much of a difference.

**Summary Of The Paper:**

The article describes the development of a traffic simulation and analysis framework named CBLab. Experiments are executed to show the effectiveness of the framework. The framework is said to perform better than the state-of-the-art. The framework provides all the tools needed for analyzing the traffic of a large city, including a simulation engine, a data engine, and a scenario builder. There are also capabilities provided that enable a user to introduce custom traffic and driver behaviors.  Although the work is much needed, but there are almost no fundamental improvements within the theoretical domain.

**Summary Of The Review:**

The paper assists the researchers in conducting research on traffic analysis using simulation. The paper is also well-written, and the results are explained nicely. Apart from one contradiction that is listed above, I could not find any discrepancies.

Nevertheless, the simulation of parallel and distributed systems is a large domain in itself, and the paper seems not to address that topic at all. There is no discussion on the type of model of simulation they have used, for example, event-driven or fixed time step, although one can imagine that they must have used event-driven simulation. Then comes the questions of parallelism, how they have managed parallel events, and whether there was a single event queue or multiple event queues for disjoint sets of simulation. All these fundamental questions are unanswered.

---

> ### Author Response · Authors · 2022-11-16
> **Response to the review**
>
> We appreciate the insightful review. We address the issues raised by points.
>
> 1. [Weakness 1: Is CBLab appropriate for ICLR?] We agree that CBLab does not include much fundamental scientific contribution but stimulates more scientific and practical methodology innovation in traffic control policies as a software infrastructure for the application of machine learning. As described in https://iclr.cc/Conferences/2023/CallForPapers, ICLR2023 welcomes “implementation issues, parallelization, software platforms, hardware”. Moreover, we can find a specific category of “Infrastructure (eg, datasets, competitions, implementations, libraries)” on the submission page of ICLR2023 on OpenReview. Our submission was encouraged by these statements and we believe that ICLR appreciates the importance of infrastructure and welcomes such submissions as CBLab.
>
> 2. [Weakness 2: The change in architecture does not make much of a difference.] As mentioned in the Introduction, although real urban traffic includes thousands of intersections and maybe over one million vehicles, existing studies on traffic policies only focus on traffic of very small scales (10-100 intersections) for two reasons. First, the efficiency and scalability of the simulator used are poor. Second, there is no data for larger-scale traffic. Our work provides an efficient simulator and enriched large-scale input data for the simulator to tackle these two problems. With our work, researchers can push the study of traffic policies forward to city-level traffic and closer to the real scenario. We believe this is significant for the field of interdisciplinary research in transportation and computer science, for it helps a step forward to the application.
>
> 3. [Clarity 1: The scalability of Cityflow] As mentioned in the review, we say that the existing simulators “are either not able to conduct a city-level simulation in a feasible time or set to prevent masses of vehicles from coming in the traffic”. Cityflow belongs to the first case, where it is unable to hold a lot of vehicles. As shown in Fig.5, Cityflow is unable to hold 100,000 vehicles, which is common in city-level traffic. In fact, it cannot even hold 30,000 vehicles. The reason is discussed in Appendix C. Therefore, we believe our description covers the case of Cityflow.
>
> 4. [Clarity 2: Page 5 -> "The stability of CBEngine is distinctly better than that of baselines" How the stability is calculated, is it a typo?] The stability is measured by the deviation of time costs of three repeated experiments, which is reported in parentheses of Table 1, such as (+-0.54).
>
> 5. [Clarity 3: There is a discussion about preprocessing, but how preprocessing is performed, the article is silent about it.] We have discussed the preprocessing in Sec 3.1, Preprocessor. As mentioned in the paper, the preprocessor removes the redundant nodes. This is implemented by detecting the distance between two nodes. In OpenStreetMap, redundant nodes are quite close to each other and sometimes even share the same location. We detect such node pairs and delete the redundant node. After that, we reconnect some roads to the rest node, which are connected to the deleted node before.
>
> Again, we appreciate the helpful review and welcome any additional comments. Please feel free to contact us if you have any further questions.

---

### Official Review · Reviewer_4dnh · 2022-10-25

**Confidence:** 3
**Correctness:** 3
**Technical Novelty And Significance:** 2
**Empirical Novelty And Significance:** 3
**Recommendation:** 6

**Clarity, Quality, Novelty And Reproducibility:**

It is a comprehensive work from the perspective of RL. The codes are provided in details.

**Strength And Weaknesses:**

### Strength:
Large-scale traffic simulation is a crucial topic and designing such a tool is useful.

### Weakness:
1. This paper lacks some key related works, such as:
Chen, H., Yang, K., Rizzo, S. G., Vantini, G., Taylor, P., Ma, X., & Chawla, S. (2020). QarSUMO: A Parallel, Congestion-optimized Traffic Simulator. GIS: Proceedings of the ACM International Symposium on Advances in Geographic Information Systems, 578–588. https://doi.org/10.1145/3397536.3422274
2. In Figure 3, why do other models divide the process into two: status and action?
3. How to ensure the authenticity of the road network (the paper didn't mention, such as the number of lanes, and connection relationship);
4. How to ensure the authenticity of traffic?



**Summary Of The Paper:**

This paper design a useful tool for scalable traffic simulation, which would benefit RL training. This system is very comprehensive, and large-scale including three modules,  CBEngine, CBData and CBScenario.

**Summary Of The Review:**

It seems to be a good tool that may facilitate more research. But I have some concerns as mentioned in weakness part.

---

> ### Author Response · Authors · 2022-11-16
> **Response to the review**
>
> We appreciate the insightful review. We address the issues raised by points.
>
> 1. [Weakness 1: Lack of some related works] We are sorry that we did not include QarSUMO in our related work and would like to add it to the new version of our paper. However, QarSUMO has not mentioned access to its code in the paper so it seems impossible to compare it with our simulator. Additionally, according to the result shown in the paper or QarSUMO, it improves the efficiency of SUMO by around 100%, while our simulator is 10 times quicker than SUMO. Therefore, our simulator may outperform QarSUMO from the perspective of efficiency and scalability.
>
> 2. [Weakness 2: Why do other models divide the process into two: status and action?] The driving decision-making is usually modeled as a Markov decision process (MDP). The modeling of status and action is square with the state and action in MDP, making it easy to comprehend and implement.
>
> 3. [Weakness 3: How to ensure the authenticity of the road network?] We extract our road networks from OpenStreetMap, the most detailed open road network database. Therefore, our road networks are of the finest granularity as possible as they can be based on open databases. We also welcome private data owners to provide more detailed road network data to our society. For details of the road networks, they include lane-level information. We follow SUMO and Cityflow, two widely-used simulators for learning traffic policies, to include three lanes in one road.
>
> 4. [Weakness 4: How to ensure the authenticity of traffic?] We ensure the authenticity of traffic from two aspects. First, the simulator supports driving model customization so users can tune the parameters of the default driving model or even rewrite a new driving model according to their needs and real-world driving logs. This is the best promise of the potential to improve authenticity among simulators for learning traffic control policies, for our baselines use fixed driving models. Second, we provide APIs to transfer trajectory data to our traffic flow data. This offers the best potential among all simulators for learning traffic control policies to achieve good plausibility, while the driving model of other simulators is fixed. Second, users can obtain a traffic flow input for the simulation with our pipeline in CBData, giving real-world trajectory data. Admittedly, we have not had a lot of trajectory data released at present because this involves a lot of privacy concerns. We believe this job should depend on the power of the open-source society. That is the reason why we make our tool completely open-sourced.
>
> Again, we appreciate the helpful review and welcome any additional comments. Please feel free to contact us if you have any further questions.

---

### Official Review · Reviewer_9smp · 2022-10-26

**Confidence:** 3
**Correctness:** 3
**Technical Novelty And Significance:** 2
**Empirical Novelty And Significance:** 3
**Recommendation:** 6

**Clarity, Quality, Novelty And Reproducibility:**

As discussed in the weaknesses before, I think the clarity and quality of the presentation could be substantially improved. I would recommend first describing the simulator together including traffic driving model, data, and benchmarks in detail before going into experiments. There are also many typos throughout that should be fixed.

Though the work provides some novelty in the sense that the simulator is faster and makes a large scale dataset of road networks available, the actual technical innovation is limited. The main improvement in simulation is an implementation detail of how things are parallelized, and no methodological improvement that leverages the simulator is presented.


**Strength And Weaknesses:**

Strengths:
* A very large scale and realistic traffic simulator could be useful for things like learning and evaluating traffic behavior models, especially in the context of reinforcement learning.
* The proposed CBData with 100 road networks across the world is an interesting dataset that may be useful for several tasks in the community.
* Experiments show the proposed simulator can handle very large scale simulations (thousands of intersections and vehicles) more efficiently than relevant previous simulators. Also, that the traffic behavior model can to some extent be tailored to a specific map through black-box optimization of parameters using GPS data.
* Extensive code and clear documentation is provided which would greatly aid reproducibility and use of the simulator.

Weaknesses:
* It is not clear until the very end of the intro that the traffic “policies” considered here are things like traffic signal control and not the driving policy that controls traffic behavior. This really should be made clearer both in the abstract and early in the intro. But I’m wondering why the simulator is only shown for high-level traffic flow tasks? As a microscopic simulator couldn’t it be directly used for learning behavior models as well?
* The paper does not mention or cite nuPlan (https://www.nuscenes.org/nuplan), l5kit (https://woven-planet.github.io/l5kit), or Carla (https://carla.org/) which are comparable traffic simulators. nuPlan and l5kit also include large scale real-world data for both maps and traffic that can be used in simulation. Though these simulators are not meant for tasks like RL, they do include detailed real-world maps along with real traffic on these maps (unlike the proposed simulator, which simply evenly distributes traffic across the map to start simulation). Moreover, at least some of them support custom traffic behavior models so it's not true that the proposed CBEngine is "the first traffic simulator to support easy-to-use customization of vehicle behavior models."
* In general, the paper does not do enough to convince that the need for 1000s of intersections and vehicles compared to 100s already supported by prior simulators. Sec 4 shows it’s possible to run baseline methods using the simulator, but I don’t think it actually “justifies the plausibility of our simulation” since metrics are only reported for the proposed simulator. It would be better to compare metrics on the proposed simulator to those in a prior simulator (like SUMO) that’s already accepted as plausible. It would also be more convincing to show an application like training an RL traffic driving model that truly requires the large-scale parallelism the simulator provides.
* The organization and presentation of the method is often confusing and unclear. For example, Sec 2.1/2.2 are high level and vague, and no details of the behavior model are given until much later in Sec 3.2.1. Also, experiments are scattered throughout the description of the methods, making them sometimes hard to follow since the full context of the method has yet to be described.
* For the simulator to be useful it’s important that traffic behavior be realistic, but currently this is not sufficiently supported. Firstly, the actual traffic model used is only very briefly described as “similar to IDM” and is only evaluated by deviation of average speed to ground truth which is a very coarse metric. A more comprehensive metric that compares the full distribution of speed and accelerations for individual vehicles would be more informative (e.g. see [BITS: Bi-level imitation for traffic simulation, Xu et al., 2022]) in evaluating realism. It’s also not clear does the simulator support more recent learned traffic behavior models (e.g. [TrafficSim, Suo et la., CVPR 2021])? Many of these are implemented in Python – is this supported by the simulators C++ interface? I would also appreciate some qualitative visualizations of the simulated traffic.
* It’s not clear what is the granularity of the road networks provided in CBData. Does this have lane-level information or is it just the coarse road network?


**Summary Of The Paper:**

This paper presents a new traffic simulator to support large scale scenarios that include thousands of intersections and vehicles. Compared to prior work, the simulator is implemented in a more efficient parallelized fashion to enable such large simulations. The paper also proposes a new dataset of 100 traffic networks processed from OpenStreet map along with benchmarks for various traffic control tasks such as traffic signal control and congestion pricing. Experiments show the simulator is more efficient than prior works and that baseline methods for traffic control tasks can indeed be run in the simulator

**Summary Of The Review:**

Overall, I can see why a large scale simulator like the one presented could be useful for several aspects of traffic modeling, but the current paper does not do enough to convince this and has several issues in terms of presentation and clarity. Moreover, I don’t think this is a great fit for ICLR considering the limited contribution on the technical/learning side.


================= After Author Response ========================

My initial review focused on the paper's connection to traffic behavior modeling and not necessarily the problem of traffic flow control, which I now understand is the focus here. The authors have clearly detailed the key differences between these two, why low-level behavior is not as important for traffic flow, and what makes simulation for traffic flow control unique. I think it's good they changed terminology to "traffic control policies" although I think "traffic flow control policies" would be even more clear.

After re-considering the work in the context of traffic flow control, I have raised my score to a borderline accept.

With the newly added information in the paper on the behavior model, I think it makes the context and organization more clear. Moreover, the added experiments showing similarity of CBEngine traffic to SUMO indicating it is reasonable enough for traffic flow applications; and the behavior can be extended with their API.

I am still concerned about the fit with ICLR seeing how the technical novelty is extremely limited; the contribution here is purely one of a simulation framework and dataset. Also, while the paper shows previous traffic flow methods can be successfully applied at new scales in the proposed simulator, there is no demonstrated empirical advantage to doing so (e.g. showing that training at a larger scale improves performance compared to smaller scale). However, previous works in this field have demonstrated the desire for operating at such city scales (Chen et al., 2020a; Zhang et al., 2022), so I do believe the proposed simulator and dataset would be a benefit to the community. In the end, I think it would be more beneficial to for this framework to be published and made available than not. And the authors have already shown a commitment to this by releasing code and detailed documentation.

---

> ### Author Response · Authors · 2022-11-16
> **Response to the review (4)**
>
> References:
>
> [1] Seung-Bae Cools, Carlos Gershenson, and Bart D’Hooghe. Self-organizing traffic lights: A realistic simulation.
>
> [2] Afshin Oroojlooy, Mohammadreza Nazari, Davood Hajinezhad, and Jorge Silva. Attendlight: Universal attention-based reinforcement learning model for traffic signal control.
>
> [3] Pravin Varaiya. Max pressure control of a network of signalized intersections.
>
> [4] Hua Wei, Guanjie Zheng, Huaxiu Yao, and Zhenhui Li. Intellilight: A reinforcement learning approach for intelligent traffic light control.
>
> [5] Libing Wu, Min Wang, Dan Wu, and Jia Wu. Dynstgat: Dynamic spatial-temporal graph attention network for traffic signal control.
>
> [6] Bingyu Xu, Yaowei Wang, Zhaozhi Wang, Huizhu Jia, and Zongqing Lu. Hierarchically and cooperatively learning traffic signal control.
>
> [7] Liang Zhang, Qiang Wu, Jun Shen, Linyuan Lü, Bo Du, and Jianqing Wu. Expression might be enough: representing pressure and demand for reinforcement learning based traffic signal control.
>
> [8] Guanjie Zheng, Yuanhao Xiong, Xinshi Zang, Jie Feng, Hua Wei, Huichu Zhang, Yong Li, Kai Xu, and Zhenhui Li. Learning phase competition for traffic signal control.
>
> [9] Guni Sharon, Michael W Levin, Josiah P Hanna, Tarun Rambha, Stephen D Boyles, and Peter Stone. Network-wide adaptive tolling for connected and automated vehicles.
>
> [10] Hamid Mirzaei, Guni Sharon, Stephen Boyles, Tony Givargis, and Peter Stone. Enhanced delta-tolling: Traffic optimization via policy gradient reinforcement learning.
>
> [11] https://www.toronto.ca/services-payments/streets-parking-transportation/traffic-management/traffic-signals-street-signs/signal-policies-and-coordination/
>
> [12] https://www.oregon.gov/ODOT/Engineering/Documents_TrafficStandards/Traffic-Signal-Policy-Guidelines.pdf

---

> > ### Comment · Reviewer_9smp · 2022-11-19
> > **Re-considering**
> >
> > I sincerely thank the authors for their detailed and informative response. It's true that much of my initial review was focused on the paper's connection to traffic behavior modeling and not necessarily the problem of traffic flow control, which I now understand is the focus here. The authors have clearly detailed the key differences between these two, why low-level behavior is not as important for traffic flow, and what makes simulation for traffic flow control unique. I think it's good they changed terminology to "traffic control policies" although I think "traffic flow control policies" would be even more clear.
> >
> > After re-considering the work in the context of traffic flow control, I will raise my score to a borderline accept. Please see my updated review for more details.

---

> > > ### Author Response · Authors · 2022-11-28
> > > **Thank you for re-considering**
> > >
> > > We sincerely appreciate your prompt review of our response. It is very commendable of the reviewer to recognize the value of traffic control policies, a field attracting relatively little attention but making sense to the public interest, and that of our work. Also, the review greatly contributes to the completeness and clarity of our paper. As for the concerns raised, we will respond to them by points.
> > >
> > > 1. [I am still concerned about the fit with ICLR seeing how the technical novelty is extremely limited; the contribution here is purely one of a simulation framework and dataset.] we must admit that the contribution of CBLab is not in proposing a novel algorithm but to stimulate the research on more advanced algorithms in the field of traffic control policies, especially in large-scale scenarios. As described in https://iclr.cc/Conferences/2023/CallForPapers , ICLR2023 welcomes “implementation issues, parallelization, software platforms, hardware”. Moreover, we can find a specific category of “Infrastructure (eg, datasets, competitions, implementations, libraries)” on the submission page of ICLR2023 on OpenReview. Our submission was encouraged by these statements and we believe that ICLR appreciates the importance of infrastructure and welcomes such submissions as CBLab.
> > >
> > > 2. [Also, while the paper shows previous traffic flow methods can be successfully applied at new scales in the proposed simulator, there is no demonstrated empirical advantage to doing so.] First, the vacancy of empirical advantage originates from the vacancy of algorithms for large-scale scenarios. As mentioned, the exploration of such algorithms is limited by the scalability of simulators, which is the bottleneck our work tries to tackle. Second, although empirical advantages are not affluent, there is experimental evidence that the traffic control problem grows very different when the scale changes. For example, algorithms proposed for the scale of dozens of intersections perform badly in the scenario of around 1,000 intersections (Table 3, Chen et al., 2020a). Also, it is a common assumption that large-scale means the problem is more difficult together with more information that can be used by the algorithm, showing the potential to improve the performance. Hence, we believe that the large-scale traffic control problem is a new challenging problem worthwhile being studied. Our work makes sense by serving as the infrastructure for such studies.
> > >
> > > 3. [I think it's good they changed terminology to "traffic control policies" although I think "traffic flow control policies" would be even more clear.] “Traffic flow control policies” is sound and clearly demonstrates its own meaning, but the concept “traffic control policies” seems to be more widely used in official documents. To avoid misconstruing similar to the one happening in the review, we decide to use the latter concept.
> > >
> > > Again, we thank the reviewer for continuing helping the improvement of our paper and raising the score. Please feel free to contact us if you have any further questions.

---

> ### Author Response · Authors · 2022-11-16
> **Response to the review (3)**
>
> 6. [Weakness 3.2 & 5.1: Plausibility] First, as mentioned in 0., the plausibility of simulators for learning traffic policies is quite different from that of simulators for learning driving behaviors, for there is no driving log in the input to refer to and all vehicles are controlled by the driving model in the simulator. As one such kind of simulator, the main contribution to the plausibility of our work does not focus on proposing a novel driving model but on developing a framework that enables users to easily migrate various driving models to the simulator. This function was absent in SUMO and Cityflow and was first implemented in CBEngine, from which CBEngine brings out the best potential to improve the plausibility. Second, our default driving model is an analytical one, following SUMO and Cityflow. As recommended by the reviewer, we conduct new experiments to show the basic plausibility of our default driving model. We compare the average speed of vehicles in CBEngine and SUMO under three of six efficiency experimental setups. The result shows a little gap between the two simulators from the perspective of vehicle speed. (The unit is m/s)
>
> | City Name            | Difference in vehicles’ average speed over 600 seconds |
> | --------------- | ---------------------------------------------- |
> | Changchun    | 1.9038 |
> | Nanchang             | 1.3983 |
> | JiNan   | 1.4609 |
>
> Furthermore, we compare the speed distribution of vehicles in 150s, 300s, and 450s of two simulators under the experimental setup of Changchun. The normalized Wasserstein distance is given as follows. (The unit is m/s)
>
> | Time Step            | Normalized Wasserstein distance between speed distributions |
> | --------------- | ---------------------------------------------- |
> | 150 | 0.0141 |
> | 300 | 0.0124       |
> | 450 | 0.0089 |
>
> Two experiments show that the default driving model in CBEngine has controllable differences between the most widely-used SUMO. More details and visualization of these two experiments would be updated in the appendix of the new version of our paper. Finally, except for macro statistics such as average speed, detailed driving behaviors are not the first concern for the tasks of learning traffic control policies. Existing studies [1-8] are mostly conducted using simulators with fixed driving behavior models and fixed parameters which are pre-assigned by the developers. To conclude, we have offered the best plausibility among simulators used for learning traffic control policies.
>
> 7. [Weakness 5.2: Does the simulator support more recent learned traffic behavior models implemented in python?] These behavior models can be connected to the simulator with a C++-Python interface.
>
> 8. [Weakness 6: Granularity of road networks] The road networks include lane-level information. We follow SUMO and Cityflow, two widely-used simulators for learning traffic control policies, to include three lanes in one road. Additionally, we make an improvement for a wider avenue by modeling it as two or more roads, which is not implemented in the dataset of SUMO and Cityflow. Since our data comes from OpenStreetMap, the most detailed open road network dataset all around the world, we have provided the most fine-grained road networks of many main cities (e. g. Shanghai) as possible as they can be based on open databases. We also welcome private data owners to provide more detailed road network data to our society.
>
> 9. [Clarity] We are sorry that our presentation is somewhere confusing and will revise it in our next version. Also, we would give a clearer description of our default driving model in the new version of our paper. However, our simulator, dataset, and benchmark are three relatively independent toolkits. It would be more confusing if we put the description of these parts together in a “method” section. Therefore, we place the experiment section after the description of every toolkit to offer a complete description for each toolkit.
> Again, we thank the helpful review and hope that our response clears the differences between learning traffic control policies and learning driving behaviors, which is essential for the comprehension of our work. We will sincerely appreciate it if you can review our work from the perspective of the field it is designed for and consider the conditions of the field accordingly. Please feel free to contact us if you have any further questions.

---

> ### Author Response · Authors · 2022-11-16
> **Response to the review (2)**
>
> 1. [Weakness 1.1: Traffic “policies” considered here are things like traffic signal control and not the driving policy that controls traffic behavior.] As mentioned in 0., traffic control policies in this paper refer to traffic control policies. Both these control policies and driving behaviors have impacts on traffic and need to be studied. Specifically, traffic control policies may obviously influence the distribution of traffic flows in practice, which makes it a very meaningful research topic.
>
> 2. [Weakness 1.2: Why the simulator is only shown for high-level traffic flow tasks?] Our simulator is designed specifically for learning large-scale traffic control policies. It would not be a problem that our simulator cannot help in learning driving behaviors, which is a definitely different task. It is noted that simulators for learning driving behaviors cannot serve as the training environment for learning traffic control policies either.
>
> 3. [Weakness 2: The paper does not mention or cite nuPlan, l5kit, or Carla, which are comparable traffic simulators, including large-scale real-world data.] nuPlan, l5kit, and Carla are all simulators and datasets for learning driving behaviors and cannot be used to learn traffic control policies. As mentioned in 0., these simulators have three key differences from simulators used as the environment to learn traffic control policies. The excellent driving behavior in these simulators can be a good reference when our future users implement their own driving behavior models. However, these simulators serve an exactly different task, not tightly related to our work and the task of learning traffic control policies. That’s the reason we do not mention or cite them in our paper. It is also noted that “large-scale” in their context means the size of trajectory data, while it refers to the scale of simulation (thousands of intersections, one million vehicles) in our context. For example, many scenarios in nuPlan only include one intersection, far from “large-scale” in the context of learning traffic control policies. Therefore, CBLab still provides the first large-scale simulation for learning traffic control policies.
>
> 4. [Weakness 3.1: the paper does not do enough to convince that the need for 1000s of intersections and vehicles compared to 100s already supported by prior simulators.] It is pointed out that larger road networks would render the solution and even the representation of the solution to traffic signal control to be more difficult (see in [State-of-art review of traffic signal control methods: challenges and opportunities, Qadri et.al, 2020]). Also, real city-level road networks are of the scale of thousands of intersections. Therefore, considering traffic control policies in road networks of such scale are left unexplored, we believe that it is in need to first conduct research to study the property of such scenarios rather than simply assuming it is the same with cases with road networks with smaller scales.
>
> 5. [Weakness 4: Organization] We are sorry that the presentation of some parts is vague and would like to correct them in our new updated version. Specifically, we will give a clear description of our default driving behavior model. Note that we implement an easy-to-use driving module customization for all vehicles. Even with the default model, users can use an API to customize the parameters of the model. Therefore, we promise the best flexibility among simulators for learning traffic control policies to adapt to different scenes.

---

> ### Author Response · Authors · 2022-11-16
> **Response to the review (1)**
>
> We appreciate the helpful review. We have noticed that several main concerns in the review can be addressed by distinguishing the key difference between the two tasks: learning traffic control policies and learning driving behaviors. The difference results in no comparability between simulators for learning traffic control policies and simulators for learning driving behaviors. Since the motivation of our work is to provide a simulator for learning traffic control policies, we are sorry that we did not provide enough details of this difference and thus caused misconstruing. To compensate for this omission, we would like to first give the background about learning traffic control policies. In the background, we will also supplement details to help distinguish two tasks and two kinds of simulators used for two tasks respectively. After that, we would address the issues raised by points:
>
> 0. [Background]
> Learning traffic control policies: In practice, transportation authorities can carry out various control policies to enhance the efficiency of urban traffic, such as tuning the traffic signal timing and conducting traffic restrictions. The goal of learning traffic control policies is to explore how to optimize these control policies according to the traffic. For instance, well-crafted traffic signal control improves transportation efficiency by optimizing traffic signals, which is validated by extensive studies [1-8]. Furthermore, congestion pricing is studied more frequently than before [9, 10]. As mentioned in the review, we have noticed that the concept of “traffic policies” may be confusing, and feel sorry for using it. We would like to supplant it with “traffic control policies” (See in [11, 12]) in the new version of our paper.
>
> **Differences between learning traffic control policies and learning driving behaviors**: First, their goals are different. Learning traffic control policies tries to improve the overall efficiency of urban traffic. Learning driving behavior aims at driving a vehicle better. Second, the inputs are different. Learning traffic control policies takes traffic flow statistics (e.g. the number of vehicles on a road) as inputs and must consider a wide range of vehicles. Learning driving behaviors concentrates on the state of several or dozens of vehicles around the target vehicle. Admittedly, details of driving behaviors do have impacts on the optimization of traffic control policies, which indicates the tendency to study the task under a more realistic traffic simulation. However, traffic flow statistics are currently representative enough for learning passable traffic control policies. Evidence for this is that current traffic control policies are trained with simulators with very simple driving behavior models [1-8]. It is good to improve the driving behavior model in this kind of simulator. Nonetheless, since the traffic flow statistics are high level and the number of vehicles is too large, the very details of driving behavior models would not play a key role in learning traffic control policies. Hence, details of driving behaviors are not the first concern.
>
> **Why simulators for learning traffic control policies and those for learning driving behaviors are not comparable**: There are distinct differences between these two kinds of simulators. First, the inputs are different. Simulators for learning traffic control policies only take the road-level route (e.g. Road A-> Road B-> Road C) of all vehicles as inputs, while those for learning driving behaviors include very detailed driving environmental information in their inputs. Second, their simulation patterns are different. Simulators for learning traffic control policies speculate various traffic with different traffic control policies under basic traffic rules so the simulated traffic may not exist in the real world. This is requisite for the exploration of training RL-based traffic control policies. On the contrary, in learning driving behaviors, simulators try to fit the fixed driving trajectory data.  Third, their APIs are different. To train a traffic control policy, the policy agents (e.g. traffic signal lights) must be open to being manipulated. Users can change the policy at any time. In the simulators for learning driving behaviors, by contrast, traffic control policies just follow the fixed data records, not open for users to operate. To conclude, simulators for learning traffic control policies infer the whole traffic roughly with different traffic control policies according to very simple inputs, while those for driving behaviors are more like replaying detailed driving scenes for certain vehicles. Simulators for learning driving behaviors cannot be used as environments in the task of learning traffic control policies. Therefore, the two kinds of simulators are not comparable with their definitely different functions.

---

### Author Response · Authors · 2022-11-17
**A new version of the paper**

We appreciate the insightful review and have updated the paper according to the advice. To make our revision more visible, we mark the modification in red. The modification focuses on the following points.

1. Correct typos in the manuscript

2. Add a detailed description of our default driving model in Appendix C

3. Add two experiments to evaluate the effectiveness & plausibility of our default driving model in Appendix C

4. Add a new citation of QarSUMO

We expect further discussion with the reviewer and welcome more helpful advice. Please feel free to contact us if you have any questions.

---

### Public Comment · ~Giseung_Park1 · 2023-03-28
**Question on Reproducibility**

Dear Authors,

Hello. I recently read this paper and carefully ran the implementation opensource code (https://github.com/CityBrainLab/CityBrainLab). Thanks for providing such a valuable work.

I have a question regarding the reproducibility of the CBLab paper. In the Traffic Signal Control scenario of CBScenario, I was unable to reproduce the results of Table 2 in the paper. Specifically, DQN performed worse than MaxPressure, even after fine-tuning the DQN algorithm based on Appendix B.3 (including self.batch_size, self.learning_start and _build_model function in dqn_agent.py).

I was wondering if the authors could explain how to reproduce the results of Table 2, where DQN outperforms the other baselines.

Thanks for the effort, and I look forward to the reply at the authors' earliest convenience.

---

### Decision · Program_Chairs · 2023-01-20

**Decision:**

Reject

**Justification For Why Not Higher Score:**

Lack of evaluation of simulated traffic datasets compared to real data.

**Justification For Why Not Lower Score:**

N/A

**Metareview: Summary, Strengths And Weaknesses:**

This paper presents a new simulator for learning traffic control policies. With careful engineering, the simulator is much more scalable than previous simulators.

Perhaps the main strength of the paper is the careful attention to scalability in the simulator, which allows it to scale to much larger simulations than existing work.

Regarding the fit to ICLR, it is indeed the case that the conference welcomes "datasets, competitions, implementations, and libraries"; the conference also welcomes papers that introduce new application domains.

One weakness of the paper is: a new benchmark should allow testing open research challenges in the field. RL is certainly an active research area within ICLR, but it is not clear from the current version of the submission what are the research challenges in RL that this simulator will test in a way that current benchmarks do not cover. Perhaps there are some such challenges, but it is hard to tell from the current version of the paper.

The main weakness of the paper is in the evaluation. The paper does a good job in evaluating the scalability of the method, and whether it can be tuned on real-world data. But the paper does not evaluate how the simulated traffic datasets that can be generated by the toolkit compare with real data (it would still be helpful even if the real-world data set is small in size and/or a private dataset), so it’s challenging to assess how useful the simulation toolkit would be. If a more rigorous evaluation has been conducted, the paper would be significantly stronger to an ML audience.